# Why Is There Always a Remnant Rib Hump Deformity after Spinal Operations in Idiopathic Scoliosis: Aetiological Implications and Recognition of the Proper Rib Level for Costoplasty

**DOI:** 10.3390/children10101697

**Published:** 2023-10-17

**Authors:** Theodoros B. Grivas, Elias Vasiliadis, George Vynichakis, Michail Chandrinos, Konstantinos Athanasopoulos, Paschalis Christodoulides

**Affiliations:** 1Department of Orthopedics & Traumatology, “Tzaneio” General Hospital of Piraeus, 18536 Piraeus, Greece; 23rd Department of Orthopaedics, School of Medicine, National and Kapodistrian University of Athens, KAT Hospital, 16541 Athens, Greece; eliasvasiliadis@yahoo.gr; 3Orthopedic Department, Gen. Hospital of Argolida—N.M. Argos, 21231 Argos, Greece; vynichakisgiorgos@gna.gr (G.V.); chandrinosmichail@gna.gr (M.C.); 4Orthopedic Department, Peristeri Medical Group, 12132 Peristeri, Greece; drathkon@yahoo.gr; 5Department of Radiology, General Hospital of Paphos, Paphos 8026, Cyprus; p.christodoulides@shso.org.cy

**Keywords:** idiopathic scoliosis, rib hump, rib cage deformity, aetiology, rib index, segmental rib index, costoplasty, surgical treatment

## Abstract

The aim of this report is to review the literature dealing with the postoperative correction of rib hump (RH) after spinal operations for adolescent idiopathic scoliosis (AIS) and its aetiological implications of hump postoperative fate for IS. Recommendations related to RH deformity for the follow-up of younger asymmetric but not scoliotic children are provided, and the concept that clinical monitoring of the chest deformity is more important than merely an initially negative radiographic examination (curve less than 10°) is underlined. Additionally, guidelines are provided based on the segments T1–T12 rib index (RI) in the existing lateral preoperative radiographs for the optimal selection of the rib level for a successfully costoplasty. This review is based on the collected articles that used either the RI method, derived from the double rib contour sign (DRCS) at the lateral spinal radiographs, or alternative methods for the assessment of the RH deformity and presented the results of the operative treatment of the scoliotic spine on RH. A total of 19 relevant articles published from 1976 to 2022 were found in PubMed. Findings: All the above articles show that not only is the hump incompletely corrected, but it recurs and worsens during the follow-up and even more intensively in skeletally immature operated scoliosis children. Conclusions and Future Directions: Surgery straightens the spine, yet the RH is corrected approximately only as much as the spinal derotation. The only way to correct the RH more is with costoplasty, which, however, is not performed in most cases for many reasons. The key reason for this phenomenon is the fact that the RH deformity (RHD) is mainly due to the asymmetric development of the ribs and much less so due to the rotation of the vertebrae in the thoracic spine. Surgery on the spine cannot limit the asymmetry of the ribs or stop the mechanism that causes their asymmetrical growth. The results presented in all the reviewed articles support the important protagonistic role of RHD on scoliogenesis, which precedes the subsequent formed spinal deformity.

## 1. Introduction

Scoliosis is a 3D condition in which the spine and thorax are deformed. In addition to spinal curves and thoracic deformity, scoliosis is frequently associated with asymmetries of the other parts of the trunk and the extremities [1]. The most common type of scoliosis is AIS. It mainly affects girls [2], and the prevalence rate varies from 2–7% according to the geography [3]. Its aetiology is unclear. Genetics, epigenetics, central nervous system (CNS), skeletal spinal growth and bone metabolism, metabolic pathways, biomechanics, and other factors are involved in scoliogenesis [4,5,6].

The deformation of the ribs is an interesting finding. It helps the physicians to diagnose scoliosis. The RHD, at the Adams bending test, is a finding that defines the number of children who will be referred to the scoliosis outpatient departments of hospitals, and it is a dominant predictor of scoliosis [7,8,9].

The aetiological implementations of rib deformity raised a question and caused a lot of discussion about whether the ribs are the first to deform, followed by the spine, or vice versa. In some studies, rib deformity is considered a secondary effect of the vertebral body tilt and rotation or coronal curvature [10,11,12,13]. The view that the deformation of the ribs is not closely related to the rotation of the vertebrae and that it precedes the deformation of the spine, at least in the thoracic curves, is mainly based on the research work of Professor J. Sevastic [14,15,16,17,18,19,20,21,22,23,24,25,26] and on other published studies [27]. Theoretically, if the deformation of the ribs were the result of vertebral rotation, then the surgical alignment and derotation of the vertebral column would postoperatively completely correct the rib cage deformity as well, which is not the case in clinical praxis. Consequently, the analysis of the published postoperative outcomes of IS scoliosis correction on the ribcage deformity will objectively provide an answer to the above question. Additionally, this review provides recommendations related to RHD for the follow-up of children with trunk asymmetry (TA), as well as for the optimal diagnosis of the rib level for a successful costoplasty based on the segments T1–T12 RI in the existing lateral spinal radiographs.

## 2. Relevant Sections

The RH expresses the thoracic deformity in IS in the transverse plane. It is of great importance to review the published reports on the correction of the RH after surgical treatment of IS by the various types of operations performed on the spine that address the spinal deformity only and do not involve the rib cage (RC). This study will provide valuable information on the relationship between the RC and the vertebral column.

In a PubMed and Google Scholar survey, we selected among the citations of the reference [28] of this review those articles that used the RI to assess and present the results of the operation of the spine and the RHD and similar articles using a different method than the RI for evaluate the RHD fate postoperatively. Nineteen pertinent published articles were found from 1976 to 2022. In these reports, the correction of the thoracic deformity in terms of any residual RHD indicates that the hump is not only insufficiently corrected compared to the spinal correction, but it also recurs during the follow-up and more intensively at the skeletally immature operated scoliotics.

## 3. Discussion

The postoperative fate of a hump after surgical treatment of IS with the following surgical techniques is presented.

### 3.1. Posterior Fusion Only

Hefti and McMaster (1983) reported on 24 children with IIS and JIS. The operation was Posterior Fusion Only before the age of 7 years. After a mean of 4.5 postoperative years, there was RH recurrence. They suggested that the recurrence of the RH was due to the postoperative increase of the vertebral rotation [29].

### 3.2. Harrington Instrumentation

Thulbourne and Gillespie (1976) reported that the RH deformity after surgery was slightly improved, and they also emphasized that RH does not correlate with Cobb angle and vertebral rotation [30]. Gains et al. (1981), using the Harrington distraction and compression system, reported on the effect of the correction of the RH for IS. The compression system contributed to the correction of RH in over 2/3 of the patients, and the correction of the rib valley was much more significant than the correction of the RH. Postoperatively, the RH correction did not correlate with spinal derotation [31].

In another prospective study with 47 patients operated by a Harrington distraction rod and posterior fusion, the changes in the RH were postoperatively assessed. It was noted that despite the surgical correction of the lateral curve, there was a progression of the rib deformity in 64% of the cases after 4 years. The RH deformity was attributed to vertebral rotation, as it was noted, “Correction of the lateral curve may thus have no effect on vertebral rotation and cannot be guaranteed to effect a permanent reduction of the RH” [32].

### 3.3. Multi Rod, Hook and Screw System—Cotrel-Dubousset Instrumentation

Delorme et al. (2000) assessed the short-term 3D changes in the shape of the RC at the apex of the curve after corrective surgery in 29 AIS patients. They reported that the used instrumentation significantly improved the three-dimensional shape of the RC, yet the changes at RH were less significant (36 and 38% at the apex and the ascent levels) than those observed at the spine level (53% in the frontal plane) [33].

### 3.4. Universal Spine System (Stratec Medical, Oberdorf, Switzerland)

Pratt et al. (2001) studied the preoperative and postoperative data from patients with right thoracic AIS. After 2 years, they stated that there was a 41% correction of the Cobb angle, 23% of the apical vertebral rotation, 47% of the apical vertebral translation, and only 22% of the maximum angle of trunk inclination. Interestingly, RH recurred regardless of age, and almost half of the initial back surface correction disappeared by 2 years. They also stated that spine and RC factors determine RH correction and suggested that costoplasty is needed for the disruption of RH deterioration [34].

### 3.5. Pedicle Screw Construct

Hwang et al. (2013) presented the results of a multicenter AIS database in a retrospective study of 99 AIS patients with a minimum of a 5-year follow-up. The thoracic scoliometer reading was 14.57 ± 5.7 preoperatively and 6.7 ± 4 degrees 5 years postoperatively, and it was non-statistically corrected, as opposed to the Cobb angle of the upper thoracic curve, lumbar curve, T5–T12 kyphosis, and Lordosis [35].

Jae-Young Hong et al. (2011) [36] studied 13 boys and 37 girls with a mean age of 14.8 (10–25) years. The type of curves was Lenke I, II, III, IV, V, and VI in 19, 6, 11, 5, 4, and 5 cases, respectively. The thoracic hump deformity index (RHD) was assessed, using an almost similar way to our RI, Figure 1, applied on CT imaging. They reported that the preoperative RHD was 0.22 ± 0.22 and the postoperative 0.12 ± 0.14. The RHD was corrected but not completely, and there was a residual RHD.

### 3.6. Pedicle Screws or Hybrid Constructs, with or without Costoplasty

Lykissas et al. (2015) conducted a multicenter study using the registry database for AIS, with a minimum follow-up of 2 years in two groups. The first group consisting of 36 patients (group I) was treated with pedicle screws, direct vertebral rotation (PSF), and no costoplasty, whereas the second group consisting of 40 patients (group II) was treated with pedicle screws, vertebral rotation, and costoplasty (PSF and Costoplasty). The rib index (RI) method was used and compared between groups. The mean Cobb angles for groups I and II were 49.7 and 49.8 degrees, respectively, preoperatively, and 10.2 and 10.9 degrees postoperatively. There was no difference in preoperative and postoperative values when comparing both groups (*p* = 0.48 and 0.96, respectively). Before spine surgery, RI for groups I and II was 1.61 and 1.80, respectively. Postoperatively, the rib indices were 1.39 for group I and 1.29 for group II. These differences were found to be statistically significant (*p* = 0.002 and 0.006, respectively). The amount of correction of RI was 0.23 and 0.51 for groups I and II, respectively. This difference was found to be significant (*p* < 0.0001). The correction percentages were 13.7% and 28.3%, respectively. This difference was also found to be significant (*p* < 0.0001). This study shows that even after PSF and costoplasty, the correction of the RH is only 51% as opposed to 23% only for PSF [37].

### 3.7. Total Pedicle Screw Instrumentation with and without En-Bloc Vertebral Column Derotation (DVR)

Mattila et al. (2013) reported on the comparison of pre- and post-op clinical and radiographical parameters in 72 patients with AIS, aged 14.7 (range 9.0–18.0) years, which included six juveniles and 66 adolescents suffering Lenke 1–4 or 6, with a minimum follow up of 2 years. It was noted that “en bloc DVR has a significant effect on radiographical spinal column derotation and may help prevent the flattening of thoracic kyphosis, but this derotation was not accompanied by better thoracic RH correction at 2-year follow-up” [38].

### 3.8. Full Transpedicular Screw Construct Compared to Hybrid Constructs

Soultanis et al. (2015) studied which of the above two constructs offers better postoperative RHD correction using the radiological method of rib index. They found that although the pre- and postoperative RI correction was statistically significant within each group, this was not the case postoperatively between the two groups. Interestingly, it appears that the RHD correction is not different, no matter what instrumentation was used. Provided that the full screw construct is powerful, the postoperative derotation and RHD correction were expected to be better than when a hybrid construct is applied, yet this is not the case in this study. It is, therefore, implied that the RHD results are more likely from the asymmetric rib development rather than from vertebral rotation, as it has been commonly believed. Another interesting implication is that the spinal deformity is the result of thoracic asymmetry, which is in line with the late Prof. John Sevastikoglou’s (Sevastik’s) thoracospinal concept [39].

### 3.9. Spine Stapling with Nitinol Staples

Haber et al. (2020) studied patients with curves of 30 degrees minimum at high risk of progress based on curve magnitude, premenarcheal status in all girls, failure of brace treatment, and skeletal immaturity. Fourteen patients with 16 curves were found eligible. All patients were followed for a minimum of 36 months. The RI was used as a surrogate for scoliometric measurement. They used the last radiographs before fusion for the three cases that went on to fusion. They found that there was no significant difference in preoperative, first postoperative, and final follow-up RI. Therefore, the RHD was not corrected properly [40].

### 3.10. Selective Thoracic Fusion (STF)

Hamzaoglu et al. (2021) conducted a study on 43 female AIS patients. In the operated thoracic curves, the mean preoperative RI was 2.18 (1.13–5.3), and the follow-up RI was 1.61 (1.14–2.84) (z = 2.886; *p* < 0.05). Therefore, even though the alteration was statistically significant, the RHD was only partially corrected [41].

### 3.11. Comparison of Three Generations Spinal Fusion Systems on RH

Full pedicle screw system;Hybrid construct;Harrington rod system.

Igoumenou et al. (2016, 2021) reported a retrospective study that compared the RHD correction using the RI in AIS patients treated with three generation spinal fusion systems. The pre- and postoperative RI and RI correction (%) were assessed. The RI correction was not significantly different among the three systems (*p* > 0.05). All three fusion systems offered significant correction of the thoracic deformity with respect to the RI. The authors noted that “it was generally assumed that the RHD results almost solely from the rotational deformity of the spine. Provided that the full transpedicular screw construct is biomechanically strong, allowing for vertebral derotation and direct transverse plane corrections, it was expected that the RI correction and RHD correction in full pedicle screw system operated patients would be significantly higher, yet this was not the case in that study”. These results imply that “the RHD does not depend only on the spinal column deformity, as it was widely considered, but that it results mainly or additionally from the thoracic deformity (asymmetry of the ribs)”. The authors considered that the scoliogeny is an issue open for discussion. These results indicate that the thoracic deformity comes first, and the spinal deformity follows. However, the RHD cannot be completely corrected even when derotation is applied. These three generation spinal fusion systems postoperatively and in follow-up corrected the RH only 30.6%, 28.2%, and 28%, respectively, compared with the preoperative RC deformity. And even though these corrections are statistically significant compared with their preoperative values, in reality, the operations failed to correct the RH completely or largely, as these results are obviously dissatisfying [42,43,44].

### 3.12. Hybrid Pedicle–Hook–Screw Technique (HS) Compared to All Pedicle Screw (AS) Techniques

Tsirikos and McMillan (2022) studied 160 AIS patients. Eighty of them were treated by a hybrid pedicle–hook–screw technique (HS), and they were compared to the other eighty who were treated with all pedicle screw (AS) techniques. RI was assessed [45]. The results for the RI assessment for the All-Screw (AS) technique produced means (range) for preoperative of 2.09 (1.4–3.7) and postoperative of 1.6 (1.1–2.4), with a correction index (%) 23.4 (0–50), and for the Hybrid Hook–Screw (HS) technique, the means were 2.1 (1.5–3.2), 1.46 (1.1–2), and (%) 30.5 (0–48), respectively, and they were not statistically significant for both techniques. Therefore, for both of these surgical techniques, postoperatively and in follow-up, the RH deformity is corrected only 23.4% and 30% compared with preoperative RC deformity, respectively. These two techniques fail to correct the RH completely or largely, as these results are obviously minimal and dissatisfying [46].

### 3.13. Assessing the Implant Density on RHD

Lertudomphonwanit et al. (2022) [45] studied 99 patients to determine whether the implant density affects the correction in posterior spinal fusion (PSF) in Lenke 1 and 2 AIS. The preoperative, immediate, and final follow-up postoperative radiographs were analyzed. No significant correlation was found between screw density and curve correction in any planes. As far as the RI is concerned, the percent RI correction was defined as (preoperative RI − postoperative RI)/(preoperative RI) × 100%. They found the preoperative RI 1.89 ± 0.29 (1.25 to 2.58), the final postoperative RI 1.6 ± 0.23 (1.1 to 2.1), and the percent RI correction 17.4 ± 10.1 (0.1 to 44.5).

Percent RI correction for LD 18.4 ± 9.1 and for HD 16.5 ± 8.1. At no costoplasty patients, they found no significant correlation of the vertebral rotation to the RI correction and to the PS density. Their results are supported by previous studies. Yang et al. [47] evaluated the vertebral rotation correction indirectly from photographs of the RH. There was no significant correlation between the implant density and the vertebral rotation, assessed by photographic parameters. Bharucha et al. [48] evaluated the thoracic angle of trunk rotation using a scoliometer, and they also found no correlation between the implant density and the thoracic angle of trunk rotation.

### 3.14. RI in Severe Progressive Main Thoracic Deformity

Pizones et al. (2016) analyzed the registered data of 113 AIS patients with severe progressive main thoracic curves. Their mean age was 14.9 ± 1.9 years, and the mean MT Cobb was 59.6 ± 11.9°. The pre-op RI was 2.5 ± 1.3, 3 ± 1.5, 2.5 ± 1 by Lenke type curves 1, 2, and 3, respectively. They stated, “We have the advantage that the RC deformity precedes that of the spine in the pathogenesis of IS in younger patients with mild deformity, so clinical RH progression could be an alert parameter preceding curve magnitude progression in recent diagnosed juvenile patients” [49].

## 4. Rib Cage Deformity—Rib Index—Aetiological Implications

Changes resulting in growing non-scoliotic subjects having truncal asymmetry (TA) and developing AIS later implicate the effect of growth on the RC deformity as a primary causal factor for the development of the forthcoming spinal deformity (IS). Grivas et al. (2007 and 2022) confirmed the role of the RC asymmetry as a leading parameter for the development of IS, concluding that in mild IS, the deformity of the thorax precedes the spinal deformity. They found that there is a lack of correlation between surface topography findings (hump) presented as TA to the Cobb angle in younger girls in mild and moderate IS and that this correlation appears later in older children [27,50].

Children with truncal asymmetry (TA) without central axis deformity, that is, spinal deformity-scoliosis, are reported in the literature (see Pruijs et al., 1992, 1995). Pruijs et al. did not conduct any study by age, even though they reported that “The conclusion is that these method (surface topography) can be applied in school screening techniques, but that they do not allow a sharp distinction between normal and pathologic cases” [51,52,53].

Nissinen et al. (1989) [7] stated that “hump size was found to be the most powerful predictor of scoliosis. Large humps were more prevalent among those children in whom scoliosis subsequently developed”. The predictive significance of baseline TA was independent of all the other determinants entered in the multifactorial logistic model. The relative risk (odds ratio) for an increase of 1 mm in hump size was 1.72 in boys and 1.55 in girls. Thus, boys with humps of 6 mm had approximately a fivefold risk of developing scoliosis as compared with boys having a symmetric trunk (hump = 0 mm) at 10.8 years.

Nissinen et al. (1993) [53] studied 896 children (430 girls and 466 boys) with no scoliosis at entry with a three-year follow-up. For the prediction of scoliosis, several anthropometric measurements were conducted. They found that the asymmetric children with an RH deformity but without a radiologically diagnosed scoliosis will develop scoliosis with an odds ratio of 1.72 in boys and 1.55 in girls [53].

In a longitudinal analysis of a case series, all children in whom progressive IS developed also had visible TA at the age of 10 years [54]. Willner (1984) [55] also reported that in the moiré photographs with the children standing in the erect position, 12% of the girls and 9% of the boys with clinically observed TA in the Adam test had very small shadow asymmetries (deviation of <1 contour line). Also, in former Malmo studies, these small TAs were not related to a lateral deviation of the spine exceeding 9° seen radiologically [7,54,55,56].

In our school screening program, approximately 30% of the younger asymmetric referred girls, less than 13 years of age, with an ATR ≥ 7°, were found to have either a straight spine or a spinal curve Cobb angle less than 10°. In this age group, the correlation between the clinical deformity presented as TA assessed with the RI method and the radiographic measurement presented as Cobb angle is not statistically significant, while in older SSS-referred girls aged 14–18 years of age, it is [27].

## 5. Conclusions and Future Directions

The authors’ opinion is that “at initiating and mild scoliosis, the patho-biomechanics are probably dissimilar from the biomechanics when the curve is severe”. Furthermore, we consider that at initiating and mild IS cases, genetics, epigenetics, and biology have the dominant/protagonistic aetiological role; however, we should not overlook the non-protagonistic role of patho-biomechanics at this stage, which become dominant later for progressive IS [50].

Based on previously published information and on findings from SSS-referred children, our evidence-based policy recommendation is that all younger children who are identified with a surface deformity using a scoliometer but without a radiographically confirmed scoliotic curve are at risk for IS development and must be followed and not discharged from the scoliosis clinics [27].

The concept that clinical monitoring of the chest deformity is more important than merely an initially negative radiographic examination (curve less than 10°) is of great significance. This is a concept that is little known to most pediatricians or specialists who do not deal with scoliosis. So often, the absence of scoliosis on radiographic examination leads to the family being reassured and monitoring to be stopped, thus missing the opportunity to make an early diagnosis and a proper treatment.

Up to now, no single gene has been recognized as the sole responsible one for scoliogenesis, and there is no conclusive evidence of whether AIS is one disease or a collection of multiple diseases. Research suggests that AIS may be influenced by genetic factors, environmental factors, and spinal growth and development. Additionally, a significant body of research indicates the presence of diverse morphological features, clinical presentations, and prognoses among AIS patients. Complexity and heterogeneity are key characteristics of AIS aetiology and phenotype, suggesting that AIS can be considered a relatively complex group of diseases. Thus, the opinion that the association of spinal deformity with rib deformity is more kind of a spectrum may be considered as an existing option. However, the spectrum is more likely inclined to disassociation of the scoliotic and rib deformity types.

During these 46 years (1976 to 2022) of surgical praxis on idiopathic scoliosis, one can find intrinsic biases and limitations in the reports, as the assessment methods used were not the same. We tried to find articles using our rib index method, which was presented internationally in 2000; therefore, this method has been used in published papers from this year on. Even though this limitation may somehow decrease the scientific value of the review, the reported outcomes in the reviewed articles of the RH correction after surgical treatment of IS mention the postoperative existence of a hump, no matter the surgical methods used.

The operation named costoplasty or thoracolpasty, or pleuroplasty, is the one that has been introduced to correct the deformity of the ribs in the RC of AIS patients [12,57,58].

The problem of the remaining RHD after the operative treatment of AIS can be sorted out using a costoplasty operation. Indeed, in operable cases of idiopathic scoliosis with an excessive hump, costoplasty is sometimes performed in addition to spinal surgery. The results are not always satisfactory, as mentioned in the literature [59], because, in some patients, there is a persistent or remaining rib deformity after the costoplasty. As a result, the patients and their families are not satisfied with the operation. The explanation of this phenomenon is provided by Erkula et al. (2003). It is difficult to predict which vertebral level is measured because the ribs slope obliquely downwards. In the case of a severe RHD, it is hard to recognize the exact vertebral level that corresponds to the maximum rib deformity. They recommend performing a scanogram or a 3D reconstruction of the spine and of the ribs, which will help to define the exact level of the rib deformity that corresponds to a certain vertebral level [60]. Their findings are very much in line with what triggered the introduction of the segments RI method [61,62,63].

The term “pattern of segmental RIs asymmetry” is used to indicate the number of rib levels, from T1–T12, with severe RC asymmetry, namely with RI value equal to or more than 1.45–1.50. The maximum segmental RI value in the pattern of increased RI values could replace the scanogram or the 3D reconstruction and will help the recognition of the exact level or levels of the rib deformity/ies corresponding to the certain vertebral level, which must be costoplasted. Thus, the patients will have less exposure to radiation, and this is considered one important benefit of using the novel introduced segmental RI method.

In conclusion, the consequential recommendations from this review are the following:The segmental RI method provides aetiological implications for IS, suggesting the primary RC deformation, which is followed by the spinal deformation at the beginning of the development of IS [61,62];All younger non-scoliotic children with TA are at risk for IS development and need to be followed [27,63];The segmental RI method is proposed to help the recognition of the rib level for the costoplasty, saving the scoliotics from further radiological investigation and exposure to radiation. This preposition may be a future research project to confirm the value of the application of the segmental RI during the preoperative investigation of the patients to recognize the correct rib level to be operated on during costoplasty [61,62].

## Figures and Tables

**Figure 1 children-10-01697-f001:**
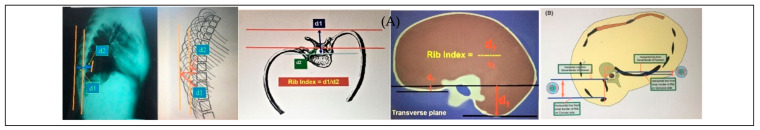
The similarity of the RI method depicted in the two middle photographs, (**A**) with this in the right photo: (**B**)-Rib hump index (RH) measured in a CD [10,36].

## Data Availability

Not applicable.

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
