# Peer review of "Why Is There Always a Remnant Rib Hump Deformity after Spinal Operations in Idiopathic Scoliosis: Aetiological Implications and Recognition of the Proper Rib Level for Costoplasty"

_children, 2023, doi:10.3390/children10101697_

Round 1
Reviewer 1 Report
Dear author, I congratulate you on your analysis.The review of the bibliography is extensive and detailed, and there are numerous useful considerations in daily practice. I agree with you that clinical monitoring of the chest deformity is more important than an initially negative radiographic examination (curve less than 10°). However, this is a concept little known to most pediatricians or specialists who do not deal with scoliosis. So often the absence of scoliosis on radiographic examination leads to the family being reassured and monitoring to be stopped, thus missing the opportunity to make an early diagnosis and a proper treatment. Could you underline this concept at the beginning, so that non-experts are immediately aware of it by reading the abstract?
Author Response
Dear reviewer. Thank you for your kind words and the very interesting suggestion to underline the concept that clinical monitoring of the chest deformity is more important than an initially negative radiographic examination (curve less than 10°). We added your recommendations in the abstract and also a new paragraph in the discussion section of the submission. See paragraphs in blue letters in the revised submission.

Reviewer 2 Report
First of all, congratulations for this paper on a very interesting and often overlooked aspect of AIS surgery.
I have some issues with the paper:
- There is an extensive use of acronyms. The first time an acronym is used, it should be reported also the extensive form, which is not always the case in the paper. For example, RI, IIS, IJS… these are all clear to us, spinal surgeons, but the paper can result troublesome for some general orthopaedic surgeons, for example.
- “Pizones et al 2016 in a cross-sectional study prospectively analyzed the registered 246 data of a consecutive cohort of 113 AIS patients with severe progressive main thoracic 247 deformity. Their mean age was 14.9 ± 1.9 years, and the mean MT Cobb was 59.6° ± 11.9°”
A 59 degrees scoliosis cohort is not a severe scoliosis cohort.
- I think that the authors should more appropriately review also the limitations of the included studies. I believe that many of the cited papers have important biases to aknowledge. Considering the non-systematic nature of the review (and therefore a considerable amount of intrinsic biases), a critical analysis of the limitations of the included paper would empower the scientific value of the review.
- I personally think that the association of scoliotic and rib deformity is more kind of a spectrum. We have certain cases in which the two deformities are paired and, therefore, a successful derotation leads to a successful rib hump correction. We have other cases where the rib deformity is primary and not secondary to the scoliotic deformity, therefore the DVR has no corrective power on the rib hump. Then, there are intermediate situations in wich the two deformities are concomitant. For example, it is common to see some unappropriately brace treated patients, in which a wrongly designed brace can induce some rib deformity. This concept should be further explained.
Author Response
First of all, congratulations for this paper on a very interesting and often overlooked aspect of AIS surgery.
Dear Reviewer, many thanks for your kind words recognizing that our submission deals with very interesting and often overlooked aspect of AIS surgery.
I have some issues with the paper:
- There is an extensive use of acronyms. The first time an acronym is used, it should be reported also the extensive form, which is not always the case in the paper. For example, RI, IIS, IJS… these are all clear to us, spinal surgeons, but the paper can result troublesome for some general orthopaedic surgeons, for example. For example, RI, IIS, IJS… these are all clear to us, spinal surgeons, but the paper can result troublesome for some general orthopaedic surgeons, for example.
Dear reviewer, thank you for your comment.
It is true that there is an extensive use of acronyms. We respected your suggested recommendation that when an acronym is used for the first time, its extensive form should also be reported, in all the cases in the paper. In order to the make the reading of the article convenient and understandable to non-familiar with scoliosis orthopaedic surgeons and physicians, we also have a list with all the abbreviations/acronyms used in this paper at the end of it, before the literature list. The repetition of the extensive forms of a definition makes the text much longer, yet an acronym initially extensively reported is the usual way to shorten the length of the paper and it is a universal policy that we followed.
“Pizones et al 2016 in a cross-sectional study prospectively analyzed the registered 246 data of a consecutive cohort of 113 AIS patients with severe progressive main thoracic deformities. Their mean age was 14.9 ± 1.9 years, and the mean MT Cobb was 59.6° ± 11.9°”. A 59 degrees scoliosis cohort is not a severe scoliosis cohort.
Dear reviewer, thank you for your comment. As far as the Cobb angle degrees related to severity scoliosis the literature is controversial. See: In literature severe scoliosis the one more than 40 degrees see, Ming-Huwi Horng, Chan-Pang Kuok, Min-Jun Fu, Chii-Jen Lin, and Yung-Nien Sun Cobb Angle Measurement of Spine from X-Ray Images Using Convolutional Neural NetworkComput Math Methods Med. 2019; 2019: 6357171. Published online 2019 Feb 19. doi: 10.1155/2019/6357171PMCID: PMC6399566PMID: 30996731, A severe curve is more than 50 degrees, see https://www.google.com/search?client=firefox-b-d&q=what+cobb+angle+is+considerd+severe+scoliosis,
At a Cobb angle measurement of 40 - 60+ degrees, this is classified as 'severe scoliosis', see. https://www.scoliosisreductioncenter.com/blog/scoliosis-degrees-of-curvature-chart
Patients with severe scoliosis (Cobb angle of 40 degrees or more), see: Kuznia AL, Hernandez AK, Lee LU. Adolescent Idiopathic Scoliosis: Common Questions and Answers. Am Fam Physician. 2020 Jan 1;101(1):19-23. PMID: 31894928.
Scoliosis Research Society recommend surgery for people with curves usually greater than 45º or 50º as these curves are considered severe and at high risk of continued worsening. Our view is that a 59 degrees’ scoliosis cohort is considered a severe scoliosis cohort.
I think that the authors should more appropriately review also the limitations of the included studies. I believe that many of the cited papers have important biases to acknowledge. Considering the non-systematic nature of the review (and therefore a considerable amount of intrinsic biases), a critical analysis of the limitations of the included paper would empower the scientific value of the review.
Dear reviewer, thank you for your comment. The review was based on only a few published articles found in PubMed, reporting the outcome of their surgical treatment on the rib cage deformity, presented as the rib hump. Nineteen relevant published articles were found from 1976 to 2022. It is obvious that in the in PubMed searching the “surgical treatment of idiopathic scoliosis” one will find 3974 articles, and only these 19 articles searching for the fate of rib hump postoperatively! Therefore, can we critically support a “non-systematic nature of the review”?
During these 46 years (1976 to 2022) of surgical praxis on Idiopathic scoliosis one can find intrinsic biases and limitations in the reports, as the used assessment methods were not the same. We tried to find articles using our rib index method, which was presented internationally at 2002. Thus, this method was used in published papers after this year. Even though this limitation may decrease somehow the scientific value of the review, the reported in the reviewed articles outcomes of rib hump correction after surgical treatment of idiopathic scoliosis, no matter the used surgical methods, mention the postoperative existence of a hump, even though the spine was aligned!
I personally think that the association of scoliotic and rib deformity is more kind of a spectrum. We have certain cases in which the two deformities are paired and, therefore, a successful derotation leads to a successful rib hump correction. We have other cases where the rib deformity is primary and not secondary to the scoliotic deformity, therefore the DVR has no corrective power on the rib hump. Then, there are intermediate situations in which the two deformities are concomitant. For example, it is common to see some inappropriately brace treated patients, in which a wrongly designed brace can induce some rib deformity. This concept should be further explained.
Dear reviewer, thank you for your comment. So far, no single gene has been identified as the sole responsible one for scoliogenesis and there is no conclusive evidence whether AIS is a single disease or a collection of multiple diseases. Research suggests that AIS may be influenced by genetic factors, environmental factors, and spinal growth and development. Additionally, a significant body of research indicates the presence of diverse morphological features, clinical presentations, and prognoses among AIS patients. Complexity and heterogeneity are key characteristics of AIS aetiology and phenotype, suggesting that AIS can be considered as a relatively complex group of diseases [67]. Thus, the opinion that the association of spinal deformity with the rib deformity is more kind of a spectrum may be considered as an existing option. However, the reported results of the included studies in this review implement that the spectrum is more likely inclined to disassociation of the scoliotic and rib deformity type.
During these 46 years (1976 to 2022) of surgical praxis on Idiopathic scoliosis one can find intrinsic biases and limitations in the reports, as the used assessment methods were not the same. We tried to find articles using our rib index method, which was presented internationally at 2000, therefore this method has been used in published papers from this year on. Even though this limitation may somehow decrease the scientific value of the review, the reported outcomes in the reviewed articles of the rib hump correction after surgical treatment of IS, mention the postoperative existence of a hump no matter the surgical methods used. We included these paragraphs in the discussion of the paper.
